# Modulation of Hepatic Functions by Chicory (*Cichorium intybus* L.) Extract: Preclinical Study in Rats [note 1]

**DOI:** 10.3390/ph16101471

**Published:** 2023-10-16

**Authors:** Lubov V. Krepkova, Alexandra N. Babenko, Svetlana V. Lemyaseva, Olga L. Saybel, Catherine M. Sherwin, Elena Y. Enioutina

**Affiliations:** 1All-Russian Institute of Medicinal and Aromatic Plants (VILAR), Moscow 113628, Russia; 2Boonshoft School of Medicine, Wright State University, Dayton Children’s Hospital, Dayton, OH 45435, USA; catherine.sherwin85@gmail.com; 3Division of Clinical Pharmacology, Department of Pediatrics, School of Medicine, University of Utah, Salt Lake City, UT 84108, USA

**Keywords:** chicory, *Cichorium intybus* L., multi-targeting properties, hypolipidemic, hypoglycemic, hepatoprotective

## Abstract

The liver is important in detoxifying organisms from xenobiotics, supporting immune functions, and metabolizing lipids and glucose. In addition, a growing number of drug-induced liver injuries and diseases associated with liver dysfunction make the development of phytodrugs targeting multiple liver functions particularly crucial. Therefore, we investigated the effects of a novel chicory extract prepared from aerial parts of the wild *Cichorium intybus* L. plant (CE) on liver enzymes and on lipid and glucose metabolism in rats with acute liver injury or hyperlipidemia. A single subcutaneous injection of mercury chloride induced an acute liver injury. Hyperlipidemia was induced by a single intraperitoneal injection of Tween-80 or by feeding rats with cholesterol and mercazolil for 28 days. Under varying regimens, the experimental rats received 100 mg/kg b.w. or 500 mg/kg b.w. of CE. CE treatment ameliorated acute liver injury by reducing liver enzyme activity, bilirubin, glucose, and lipid levels. Treatment of hyperlipidemic rats with CE effectively reduced serum lipid and glucose levels. The data obtained in this study suggest that chicory-based phytodrugs may be used to effectively treat acute liver injury and for the prophylaxis or treatment of diseases such as hyperlipidemia, type 2 diabetes, and metabolic syndrome. Clinical trials are needed to prove the effectiveness of chicory extract in human patients.

## 1. Introduction

The liver is a critical organ supporting multiple functions, including nutrient metabolism, xenobiotic compound detoxification, immune and endocrine functions, and lipid and cholesterol homeostasis [1]. Dysregulation of lipid metabolism usually manifests in increased levels of triglycerides and total cholesterol and low levels of high-density lipoprotein (HDL), which is one of the hallmarks of metabolic syndrome [2]. Obesity, elevated fasting glucose levels, and high blood pressure are other diagnostic criteria for metabolic syndrome. Since the 1990s, when Dr. Reaven introduced this syndrome to the medical community, the number of people with metabolic syndrome has increased significantly [2]. According to the NIH National Heart, Lung, and Blood Institute, “1 in 3 adults have metabolic syndrome” in the U.S. [3]. The pathophysiology of the syndrome is not well understood. It appears that increased calorie uptake, low physical activity, and, potentially, genetic factors may lead to chronic inflammation, insulin resistance, and alterations of the renin–angiotensinogen system [2]. The downstream effects lead to the development of dyslipidemia, type 2 diabetes, hypertension, and cardiovascular diseases.

A recent systematic analysis of the relationship between metabolic syndrome and liver function revealed an increased risk (49–86%) of developing liver-associated diseases such as hepatic fibrosis, acute liver failure, and cirrhosis [4]. Patients’ extensive use of drugs, herbal remedies, and dietary supplements in the modern world could also significantly affect liver functions, leading to Drug-Induced Liver Injury (DILI) [5]. Most liver injuries associated with prescription or non-prescription drugs are due to an acetaminophen (paracetamol) overdose. An overdose of acetaminophen may lead to acute liver failure and death. The excessive use of medicinal herbs and dietary supplements may also cause DILI [6]. The basic mechanisms leading to DILI are associated with the cytolysis of hepatocytes and induced inflammation by recruited immune cells [7,8]. Tyrosine kinase inhibitors, for example, Crizotinib, exhibit liver toxicity. This side effect of chemotherapeutic drugs is well known, and patients are carefully monitored for possible liver injury. However, the concomitant uncontrolled use of medicinal herbs with prescription drugs may potentially enhance liver distress [9,10]. Therefore, pharmacologists must develop effective medicines for the prophylaxis and treatment of liver dysfunction.

Nowadays, patients prefer to use herbal medicines and natural products to support their well-being. It has been reported that several constituents isolated from medicinal plants have hepatoprotective properties [5]. Among them are phenolic components (e.g., salvianolic acid, resveratrol, and curcumin), flavonoids (e.g., hesperetin, quercetin, and luteolin), and terpenoids (e.g., ginsenosides and betulin). The phenolic compounds present in the leaves of chicory (*Cichorium intybus* L., *C. intybus*) have multiple health benefits, including hepatoprotective, antioxidant, antidiabetic, and antimicrobial properties [11]. In addition, chicory is a part of a multi-component phytodrug (Liv-52) marketed by the Himalaya Drug Company since 1955 as a phytodrug intended to treat various liver diseases [12,13,14].

The chicory plant contains vitamins (e.g., vitamins C and A, thiamin, riboflavin, niacin, and pantothenic acid), minerals (e.g., Ca, Mg, Fe, P, K), and amino acids (e.g., tryptophan, leucine, isoleucine, valine, and arginine) [11,15]. Extracts from the aerial part of chicory demonstrated antioxidant and antimicrobial properties [16]. Methanolic and aqueous extracts of the aerial part showed broad-spectrum antimicrobial (e.g., *Staphylococcus* spp., *Pseudomonas aeruginosa*, and *Klebsiella pneumoniae*) and antifungal (e.g., *Candida albicans*) activity. The plant roots contain high amounts of inulin, a water-soluble polysaccharide (40–60%). Inulin may regulate low-density lipoprotein and triglyceride levels and support the intestinal microbiome [15]. Therefore, chicory-based drugs may represent a promising approach for preventing acute liver injury by hepatotoxic drugs and for the prophylaxis of metabolic-syndrome-associated diseases.

A few years ago, the All-Russian Institute of Medicinal and Aromatic Plants (VILAR) developed a novel highly purified extract from the aerial part of wild chicory collected during the flowering stage (CE). The ultimate goal was to obtain an extract with the highest concentration of polyphenolic constituents. The extraction of phenolic compounds is usually performed using 60–80% ethanol or methanol [16,17,18]. CE was obtained via the triple extraction of dried plant material with 70% *v*/*v* ethyl alcohol. Our initial study determined that this extraction method provides ~9.2–9.3% *w*/*w* of phenolic constituents calculated as chicoric acid (CA) [19]. The same study reported that CE possesses hepatoprotective activity in the model of acute hepatitis induced by tetrachloromethane (CCl4). A preclinical analysis of the extract demonstrated its low toxicity following oral and intraperitoneal administration [20].

The main objective of this study was to investigate whether CE improves liver functions by normalizing lipid and glucose metabolism in experimental animal models of hyperlipidemia. We also confirmed previous observations that CE can effectively prevent acute liver injury.

## 2. Results

### 2.1. Chemical Composition of CE

Content analysis of the extract showed that oxycoumarins, derivatives of hydroxycinnamic acids, and flavonoids represented the main phenolic compounds in CE. More than 20 compounds were identified in the extract (Appendix A). The CE’s predominant compounds were cichoriin (1.44 ± 0.05% *w*/*w*), esculetin (0.44 ± 0.14% *w*/*w*), CA (2.90 ± 0.09% *w*/*w*), chlorogenic acid (1.77 ± 0.06% *w*/*w*), and caftaric acid (0.20 ± 0.01% *w*/*w*) (Appendix A). The study extract contained 9.20 + 0.46% *w*/*w* of phenolic constituents, calculated as CA (Figure 1).

### 2.2. Hepatoprotective Properties of CE

The hepatotoxic effect of the subcutaneous (SQ) administration of HgCl_2_ was evidenced by a decrease in body weight; an increase in the relative liver weight; increased activity of γ-glutamyl transferase (GGT), alkaline phosphatase (ALP), aspartate aminotransferase (AST), and alanine aminotransferase (ALT); and destruction of lipid and glucose metabolism compared to the untreated control (Table 1). Serum levels of total proteins, triglyceride, total cholesterol, total bilirubin, and glucose significantly increased in the HgCl_2_-treated group compared with the untreated control. The body weight in the HgCl_2_-treated group dropped by 15% on day 10 after treatment initiation compared to the untreated control group and recovered to almost the body weight of naïve controls by day 21. The relative liver weight in rats receiving SQ HgCl_2_ increased by ~30% compared to that in the untreated control group on day 21 after treatment initiation (*p* < 0.05).

The body weights of animals receiving HgCl_2_ treatment followed by treatment with CE at doses of 100 mg/kg per body weight (mg/kg) or 500 mg/kg also fell by 11% and 8% on day 10 compared to the untreated control and recovered to weights comparable to those of the untreated controls on day 21 after treatment initiation. The relative liver weight of animals receiving CE was higher than the relative liver weight of untreated control animals but significantly lower than that of rats receiving HgCl_2_ only (Table 1).

The administration of CE in doses of 100 and 500 mg/kg to HgCl_2_-treated animals significantly reduced serum levels of GGT and ALT, but a statistically significant reduction in ALP and AST was observed only in animals receiving 500 mg/kg of CE (Table 1). Rats with acute liver injury treated with either 100 mg/kg or 500 mg/kg of CE also demonstrated a significant reduction in the levels of triglyceride, total cholesterol, total bilirubin, and glucose compared to HgCl_2_-treated animals (*p* < 0.05) (Table 1).

### 2.3. Hypolipidemic and Hypoglycemic Effects of CE on the Tween-80 Model of Hyperlipidemia in Rats

A single intraperitoneal (i.p.) administration of Tween-80 caused a significant increase in the levels of LDL (44%), triglycerides (40%), and glucose (14%) and a decrease in the levels of HDL (22%) in the serum of Tween-80-treated rats compared to naïve controls (Table 2). Total cholesterol levels increased by ~13%, but the differences were not statistically significant.

The prophylactic administration of CE at doses of 100 and 500 mg/kg for 14 days before the Tween-80 injection significantly reduced the levels of LDL by 32% and 26%, triglycerides by 37% and 32%, and glucose by 20% and 25%, respectively, compared to those of Tween-80-treated rats (Table 2). CE pretreatment also increased the serum HDL levels in rats by 14% (100 mg/kg) and 18% (500 mg/kg) compared to those in Tween-80-treated rats (*p* < 0.05, Table 2).

### 2.4. Hypolipidemic and Hypoglycemic Effects of CE on the Model of Alimentary Hyperlipidemia

The administration of cholesterol–mercazolil to experimental rats resulted in a statistically significant increase in the total cholesterol and triglyceride levels compared to those of the untreated control (Table 3). Rats with alimentary hyperlipidemia demonstrated a slight increase in HDL, LDL, and glucose levels, but the differences were not statistically significant. Animals receiving cholesterol–mercazolil gained weight slightly faster than the untreated control group, but the differences were not statistically significant. Based on our observations, rats on the high-cholesterol diet gained significantly more weight than did untreated control animals 80–90 days after transfer to the high-cholesterol diet.

The oral administration of CE to rats with alimentary hyperlipidemia at doses of 100 mg/kg and 500 mg/kg significantly reduced the serum levels of triglycerides by 30% and 21%, respectively (*p* < 0.05); reduced the levels of LDL by 28% and 26%, respectively; and moderately reduced the levels of total cholesterol by 14% and 16%, respectively, compared to those in animals with alimentary hyperlipidemia (Table 3). Additionally, animals treated with 500 mg/kg of CE had low body weight on day 28 of treatment initiation compared to rats with alimentary hyperlipidemia, and the differences were statistically significant (Table 3).

The liver histology of the control rats is presented in Figure 2A. The histological evaluation of livers from rats receiving the cholesterol–mercazolil treatment revealed that the treatment resulted in hyaline-droplet degeneration of hepatocytes (Figure 2B). Treatment of these rats with CE 100 mg/kg and 500 mg/kg led to a more normal liver appearance (Figure 2C,D). The liver capsule consists of densely packed connective tissue fibers. The hepatic beams are radially oriented, the lobules are formed correctly, the hepatocytes have a pale eosinophilic cytoplasm, and the nuclei are preserved. The architectonics of the vessels are normal. The endothelium of the vessels of the portal tract is not changed; the bile ducts are lined with cuboidal epithelium.

## 3. Discussion

This study supports previously published observations that CE demonstrates hepatoprotective properties. Previously, we showed that CE improves the liver function of rats with acute liver injury induced by SQ injection of CCl_4_ [19]. The current study confirmed the hepatoprotective properties of CE in another model: acute intoxication with HgCl_2_.

The study results demonstrate that treatment of rats with CE restored body weight and reduced relative liver weight, total cholesterol levels, triglycerides, glucose, and hepatic enzymes in the model of acute liver injury. *C. intybus* is a known plant with hepatoprotective properties [21,22]. For example, chicory syrup treatment decreased lipid peroxidation and improved liver histology in weanling rats exposed to deltamethrin [23]. The treatment of rats with obstructive cholestasis with 70% ethanolic extract of chicory reduced bilirubin and liver enzyme (ALT, AST, and ALP) levels [24]. Additionally, this treatment significantly decreased serum levels of TNFα and nitric oxide. Chicory extract administered to rats with thioacetamide-induced liver cirrhosis was able to normalize serum liver enzyme activity, albumin, and bilirubin levels [25]. This treatment reduced oxidative stress levels and IL-6 production by regulating AMP-activated protein kinase signaling. 6-Gingerol, a main constituent of ginger, attenuated liver injury induced by a well-known hepatocarcinogen, diethylnitrosamine [26]. The authors ascribe the hepatoprotective properties of 6-gingerol to its ability to reduce oxidative stress and TNFα protein levels.

Hepatoprotective compounds can be classified based on their mechanisms of action: antioxidants, compounds capable of regenerating hepatocytic membranes, and stimulators of regeneration of hepatic parenchyma [27]. The mechanisms of chicory’s hepatoprotective properties can be mainly attributed to its antioxidant activity. The hepatoprotective effects of chicory extracts have been attributed to the presence of phenolic compounds, specifically, chicoric acid, and its antioxidant and anti-inflammatory properties [19,28]. The hepatoprotective effects of another polyphenolic compound, kaempferol, are also ascribed to its antioxidant and anti-inflammatory properties [29,30].

The presented data showed CE’s pronounced hypolipidemic effect on two hyperlipidemia models induced by single i.p. Tween-80 administration or oral administration of cholesterol and mercazolil for 28 days. Currently, various experimental models of hyperlipidemia are used to study the hypolipidemic properties of drug candidates. One such model is experimental hypercholesterolemia induced by feeding animals a diet with excess cholesterol and simultaneous treatment with a drug interfering with thyroid peroxidase (e.g., mercazolil). The metabolism of lipoproteins in rodents (e.g., mice and rats) is different to that in humans. Rodents lack cholesteryl ester transfer protein, which is critical in human lipoprotein metabolism [31]. Inadequate enzyme activity may lead to increased levels of HDL and LDL. The absence of cholesteryl ester transfer protein explains the difficulties with the induction of hyperlipidemia in rats. It has been demonstrated that hypothyroid animals have increased LDL and very-low-density lipoproteins [32]. Triiodothyronine (T3) regulates hepatic cholesterol metabolism, specifically, cholesterol 7-alpha-hydroxylase, an enzyme involved in bile acid synthesis [33]. This explains the use of mercazolil in the rodent models of hyperlipidemia. The use of the “mercazolil model” has another advantage: it models conditions of hyperlipidemia in patients with hypothyroidism and suggests that the test drug would be effective in this category of patients.

Interestingly, the administration of CE to healthy rats with normal lipid levels for 90 days significantly decreased serum levels of low-density lipoproteins and glucose (data not shown). Keshk et al. reported that whole chicory plant powder administered to hyperlipidemic rats reduced total cholesterol, HDL, LDL, ALT, and AST levels [34]. The authors concluded that chicory and other herbal medicines could be an inexpensive alternative for treating dyslipidemia in humans.

Our study also found that CE reduced serum glucose levels in hyperlipidemic rats. Several studies have demonstrated the effectiveness of chicory extracts in animal models of type 2 diabetes. For example, one week of oral treatment with 80% ethanol/water extract of *C. intybus* in diabetic mice induced by nicotinamide–streptozotocin resulted in significantly reduced serum glucose levels [35]. Chicory extract enriched for caffeoylquinic acid ameliorated dyslipidemia and hyperglycemia by potentially reducing oxidative stress in rats fed a high-cholesterol/fructose diet [36]. A recent study demonstrated that chicory extract and a mixture of chicoric and chlorogenic acids improved glucose tolerance and normalized glucose levels in streptozotocin-treated diabetic rats [28]. It has been reported that extracts from dandelion and burdock enriched with chicoric and chlorogenic acids cause insulin resistance in rats on a high-fructose diet (experimental model of metabolic syndrome) [29].

The ability of chicory extracts to treat diabetes can be attributed to their antioxidant properties [30]. Indeed, both chicoric and chlorogenic acids demonstrated significant antioxidant properties, but only chicory extract significantly reduced glucose levels in streptozotocin-treated diabetic rats [30]. Therefore, the antidiabetic properties of chicory cannot be entirely attributed to the antioxidant activity of its bioactive compounds.

The anti-inflammatory activities of the bioactive compounds present in chicory extract may potentiate the hypoglycemic activity of the extract. White adipose tissues are a major source of inflammatory cytokines (e.g., TNFα, IL1, IL6, and others) and play a critical role in the development of type 2 diabetes [37].

Chicory extract also decreased serum glucose levels and normalized glucose tolerance test results in mice fed a high-fat diet for 6 weeks [38]. The authors suggest that one of the potential mechanisms of the hypoglycemic effects of chicory extract is its ability to reduce serum IL1β levels through inhibition of NLRP3 inflammasome activation. Furthermore, the caffeic acid moiety (e.g., caffeoylquinic, chlorogenic, 3-caffeoylquinic, caffeic, and quinic acids) is responsible for the hypoglycemic effect by reducing the activities of glucose-6-phosphatase and phosphoenolpyruvate carboxykinase, enzymes regulating glycogenesis [39].

The number of people with metabolic syndrome is rapidly increasing, and as a result, we see a significant increase in the number of patients with type 2 diabetes and cardiovascular diseases [2]. It has been believed that abdominal obesity is one of the earliest symptoms of metabolic syndrome [40]. The next stage of metabolic syndrome is the development of hepatic and pancreatic steatosis [40]. Increased lipid and glucose levels are biomarkers of metabolic syndrome [2]. Another key biomarker of metabolic syndrome is the development of insulin resistance [40]. Our data and the above studies demonstrate that chicory or chicory bioactive compounds possess hypolipidemic and hypoglycemic properties that may reduce insulin resistance. Therefore, chicory-based phytodrugs may be used to prevent and treat metabolic syndrome.

A recent publication by Pouille et al. demonstrated that, similarly to the aerial part of chicory, its roots possess anti-inflammatory, antioxidant, hypolipidemic, and hypoglycemic properties [41]. The authors attributed the observed chicory root extract activity to the presence of fructose. Chlorogenic acids and sesquiterpene lactones found at smaller concentrations in the root extract significantly suppressed IL1β and IL8 production by U937-differentiated macrophages. Chlorogenic acids are esters formed between caffeic acid and the 3-hydroxyl of L-quinic acid, and they include 4-O-caffeoylquinic acid and 5-O-caffeoylquinic acid [42]. These bioactive compounds were also identified in CE. This might suggest that chlorogenic acids are one of the bioactive compounds affecting inflammatory cytokine production by immune cells.

Interestingly, bioactive compounds from chicory demonstrated beneficial effects on other organ functions. For example, CA showed neuroprotective properties in the mouse model of Parkinson’s disease via dopaminergic neuron survival and suppression of glial-mediated neuroinflammation [43]. Additionally, the treatment of mice with CA reduced IFN-γ and IL17 protein expression in the serum, striatum, spleen, and colon. CA also attenuated renal tubular damage in obese mice by reducing mitochondrial damage and oxidative stress [44]. The same study showed that treating obese mice with CA reduced body weight and lipid and glucose levels—symptoms associated with metabolic syndrome in humans [44].

In summary, chicory and its bioactive compounds have multiple beneficial effects on liver, brain, and kidney functions. In this study, CE demonstrated simultaneously hepatoprotective, hypolipidemic, and hypoglycemic properties (Figure 3). CE may help treat acute xenobiotic-induced liver injury (Figure 3). CE can also be used as a hypoglycemic and hypolipidemic phytodrug (Figure 3). Based on the dysregulation of lipid and carbohydrate metabolism in patients with metabolic syndrome, chicory extracts should effectively prevent and treat metabolic syndrome (Figure 3). Based on the reported potential mechanisms responsible for the hepatoprotective, hypolipidemic, and hypoglycemic properties of chicory plant, the potential mechanisms of CE’s effects on liver functions are likely to be its antioxidant and anti-inflammatory properties. Our research group is conducting experiments to confirm this statement. Unfortunately, little is known about the effects of chicory extracts on human organisms. Therefore, it is imperative that clinical trials are designed to evaluate chicory extracts and determine their efficacy and toxicity in humans.

## 4. Materials and Methods

### 4.1. Phytodrug

The highly purified dry extract (CE) was obtained from the aerial part of wild chicory during the flowering stage. CE was developed at the All-Russian Research Institute of Medicinal and Aromatic Plants (VILAR). The aerial part of the chicory (*C. intybus*) was harvested in the Ryazan region in 2020–2021 according to the VILAR recommendations [45] and the International Standard for Sustainable Wild Collection of Medicinal and Aromatic Plants [46].

The extract was obtained by threefold extraction of dried plant material with 70% *v*/*v* ethyl alcohol at 50 ± 5 °C. After extraction, the liquid components were separated from the solid biomass via vacuum filtration at 50 ± 5 °C. The aqueous–alcoholic extract was concentrated, treated with dichloroethane (Chemical Point UG, Oberhaching, Germany) to separate aqueous part of extract, and dried. The resulting dry extract was crushed, sieved, and used for the phenolic compound profiling and for the evaluation of hepatoprotective, hypolipidemic and hypoglycemic properties of CE.

### 4.2. Profiling of Polyphenolic Compounds Present in CE

Polyphenolic compounds were identified in a similar way to that described by Chawech et al., with some modifications [19,47]. Briefly, polyphenolic compounds from CE were profiled using an LCMS-8040 high-performance liquid chromatography (HPLC) system (Shimadzu, Kyoto, Japan), including a Nexera HPLC chromatograph and triple quadrupole detector with an electrospray ionization source (liquid chromatography with tandem mass spectrometry, LC-MS/MS). The compounds were separated on a Luna 5 µm C18 100 Å (250 × 4.6 mm) column (Phenomenex, Torrance, CA, USA). The mobile phases (A and B) comprised 0.2% formic acid solution (A) and acetonitrile (B), respectively. The elution program was set as follows: 0–20 min, linear gradient from 10% eluent B; 20–30 min, linear gradient from 10 to 25% eluent B; 30–40 min, linear gradient from 40% eluent B; 40–44 min, linear gradient from 60% eluent B; 44–48 min, linear gradient from 80% eluent B; 48–60 min, linear gradient from 10% eluent B. The flow rate was maintained at 1 mL/min. The CE (0.01 g) was first dissolved in 25 mL of ethyl alcohol 70% (*w*/*v*). The extract filtrate (10 μL) was injected into the HPLC to separate phenolic compounds. Peaks were identified in negative and positive ionization modes. The mass spectra were obtained in the mass-to-charge range of 100–1000. Data were analyzed using LabSolutions (Shimadzu, Version 5.3).

### 4.3. Quantification of Total and Dominant Polyphenols in CE

The detailed protocol of quantification of the total and dominant polyphenols was described by Thomas et al. [19]. The total content of phenolic compounds was determined using a UV-1800 spectrophotometer (Shimadzu, Japan) at a wavelength of 327 nm. The content of phenolic compounds was calculated as CA.

The quantification of the dominant polyphenolic compounds was performed using a Prominence-I LC-2030C 3D HPLC system (Shimadzu, Japan) connected to a photodiode array detector. The column and analysis conditions were the same as those applied in the liquid chromatography–mass spectrometry (LC-MS/MS) analysis, with a 10 µL injection volume of samples. Peak identification was performed at 330 nm. Data analysis was achieved using LabSolutions (Version 5.73). Each dominant compound was quantified by constructing a standard curve with different external standard concentrations, including cichoriin, esculetin, chicoric acid, chlorogenic acid, and caftaric acid.

### 4.4. Animals

The study of CE’s hepatoprotective and hypolipidemic effects was carried out per the “Guidelines for conducting preclinical studies of drugs” implemented by the Ministry of Public Health of the Russian Federation [27]. The study protocol was approved by the VILAR Ethical Committee on Animal Experimentation and carried out per the “European Convention for the Protection of Vertebrate Animals Used for Experimental and Other Scientific Purposes (ETS 123). Strasbourg, 1986” [48,49].

Wistar male rats were obtained from the animal nursery “Andreevka”, a branch of FSBIS of the Scientific Center of Biomedical Technologies of the Medical and Biological Agency, Moscow, Russia. Animals were housed in groups of 4–5 rats in a plastic cage. Animals were kept under controlled environmental conditions of light (12 h light cycle), temperature (20–22 °C), and humidity (40–60%) with free access to standard chow for laboratory rats (Laboratorkorm LLC, Moscow, Russia) and water.

Rats were lightly anesthetized with isoflurane using a drop jar method to reduce stress following short procedures (e.g., a blood draw). At the end of each experiment, anesthetized by isoflurane, animals were euthanized by CO_2_ inhalation at a chamber replacement range (CRR) of 30% [50].

### 4.5. Analysis of CE’s Hepatoprotective Effect on the Model of Acute Liver Injury in Rats

The hepatoprotective activity of CE was studied in a model of acute liver injury induced by a single SQ injection of mercury chloride (HgCl_2_, 3 mg/kg) into male rats [50]. Male rats were randomly assigned to four groups (10 rats/group). Rats receiving a single SQ saline injection were used as an untreated control. Animals receiving a single SQ injection of HgCl_2_, with 3 mg/kg H_2_O orally 60 min before and 20 days after HgCl_2_ treatment, were used as positive controls for liver injury. The CE stock solutions were prepared daily by dissolving the dry extract in tap water. CE was administered orally by gavage in 100 and 500 mg/kg doses 60 min before HgCl_2_ administration and for 20 days following the treatment. On day 21, all animals were weighed and anesthetized with isoflurane, and peripheral blood was collected (~1.0 mL) from the lateral tail vein.

The collected blood was spun down at 1000 g for 15 min at room temperature, and serum was collected. Several biochemical parameters were analyzed using a URIT-8030 Automated Chemistry Analyzer (China Medial Devices, North Andover, MA, USA): total proteins, triglyceride, total cholesterol, total bilirubin, glucose, γ-glutamyl transferase (GGT), alkaline phosphatase (ALP), aspartate aminotransferase (AST), and alanine aminotransferase (ALT).

### 4.6. Analysis of CE’s Hypolipidemic Effect on the Model of Hyperlipidemia Caused by a Single Intraperitoneal Injection of Tween-80

Hyperlipidemia was induced by a single i.p. administration of Tween-80 [27,51]. The experiment was conducted on 32 Wistar male rats with an initial body weight of 400–500 g. The experimental animals were randomly assigned to four groups of 8 rats per group. Animals receiving i.p. saline and oral tap water were used as a naïve control (group 1); animals receiving a single i.p. injection of Tween-80 (Chemical Point UG, Germany) at a dose of 200 mg/100 g of body weight and tap water orally were used as a positive control (group 2, hyperlipidemia model); and group 3 and 4 rats were treated with CE for 14 days prior to the i.p. Tween-80 injections. The CE was administered by gavage at 100 and 500 mg/kg doses. The CE solutions were prepared daily by dissolving the dry extract in tap water. Eight hours after Tween-80 administration, peripheral blood samples were collected from the tail vein of isoflurane-anesthetized rats. Serum was obtained by centrifugation of blood samples, and biochemical parameters were analyzed (total cholesterol, LDL HDL, triglycerides, and glucose) using a URIT-8030 Automated Chemistry Analyzer (China Medial Devices, Beijing, China).

### 4.7. Study of the Lipid-Lowering Effect on a Model of Alimentary Hyperlipidemia

Alimentary hyperlipidemia in rats was induced by daily intragastric administration (by gavage) of cholesterol and mercazolil for 28 days [52]. Thirty-two male rats were randomly assigned to four groups of eight rats per group. Group 1 received daily tap water by gavage and was used as a naïve control; group 2 received daily cholesterol (Chemical Point UG, Germany) and mercazolil (Akrikhin, Moscow, Russia) for 28 days by gavage and was used as a positive control; and groups 3 and 4, in addition to the daily administration of cholesterol and mercazolil, were also treated with CE at doses of 100 and 500 mg/kg, respectively, 1 h prior to cholesterol–mercazolil administration for 28 days. CE solutions were prepared fresh prior to administration by dissolving dry extract in tap water. Cholesterol–mercazolil solution was prepared from 30 mg of cholesterol and 1 mg of mercazolil per 100 g of body weight, mixed with warm vegetable oil, and cooled down for animal treatment.

On day 29 after treatment initiation, rats were weighed, and peripheral blood samples were collected from the tail vein of anesthetized rats. The serum was obtained by 1000× *g* centrifugation for 15 min. Serum samples were analyzed for total cholesterol, LDL, HDL, triglyceride, and glucose levels using a URIT-8030 Automated Chemistry Analyzer (China Medial Devices, China).

Then, the animals were euthanized by CO_2_ inhalation. Livers were collected, and the relative liver weight was calculated using the following formula: Relative liver weight = [liver weight/body weight] × 100.

The livers from five rats in each group were fixed in 10% formalin; histological sections were then made, stained with hematoxylin and eosin, and examined using a light microscope at a magnification of ×100.

### 4.8. Statistical Analysis

All data obtained from experimental and control animals were included in the analysis. Statistical analysis of the obtained results was conducted using TIBCO Statistica, version 13 (TIBCO Software Inc., Palo Alto, CA, USA). The difference between the control and experimental groups was considered statistically significant at *p* < 0.05.

## 5. Conclusions

This study demonstrated that chicory (*Cichorium intybus* L.) extract made from aerial parts of the plant possesses simultaneously hepatoprotective, hypolipidemic, and hypoglycemic properties. The hepatoprotective properties of CE previously reported in the liver injury model induced by CCl_4_ were confirmed using the rat model of acute liver injury induced by heavy metals. Animals exposed to HgCl_2_ and treated with CE showed reduced serum levels of GGT, ALT, ALP, AST, triglyceride, and total cholesterol. The treatment with CE reduced levels of lipids—specifically, HDL, LDL, triglycerides, and total cholesterol—and glucose in hyperlipidemic rats. To our knowledge, this is the first demonstration of single extract that can alter multiple liver functions.

The hepatoprotective, hypolipidemic, and hypoglycemic properties of CE were investigated in models of acute liver injury and hyperlipidemia in rats and cannot be generalized to humans. Clinical studies are needed to confirm these findings. The exact mechanisms by which CE supports liver function are unknown. Our group is currently conducting experiments to shed light on the mechanisms responsible for the extract’s activities.

## Figures and Tables

**Figure 1 pharmaceuticals-16-01471-f001:**
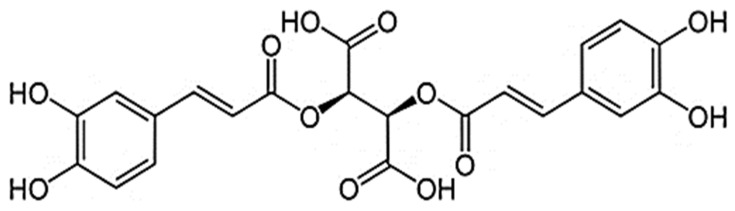
Chicoric acid.

**Figure 2 pharmaceuticals-16-01471-f002:**
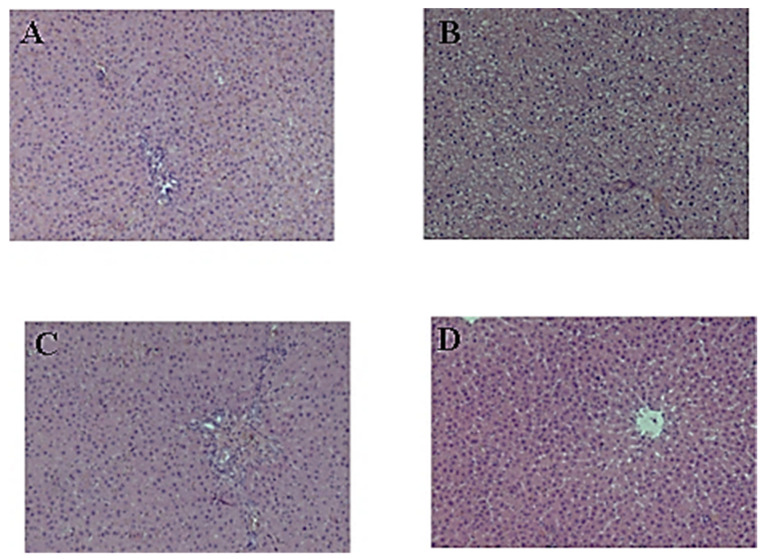
Effect of CE on the liver structure: (**A**) liver histology of untreated control rats; (**B**) liver histology of rats 29 days after cholesterol–mercazolil treatment; (**C**) liver histology 29 days after cholesterol–mercazolil treatment and 100 mg/kg CE; (**D**) liver histology 29 days after cholesterol–mercazolil treatment and 500 mg/kg CE. Hematoxylin and eosin staining, magnification ×100.

**Figure 3 pharmaceuticals-16-01471-f003:**
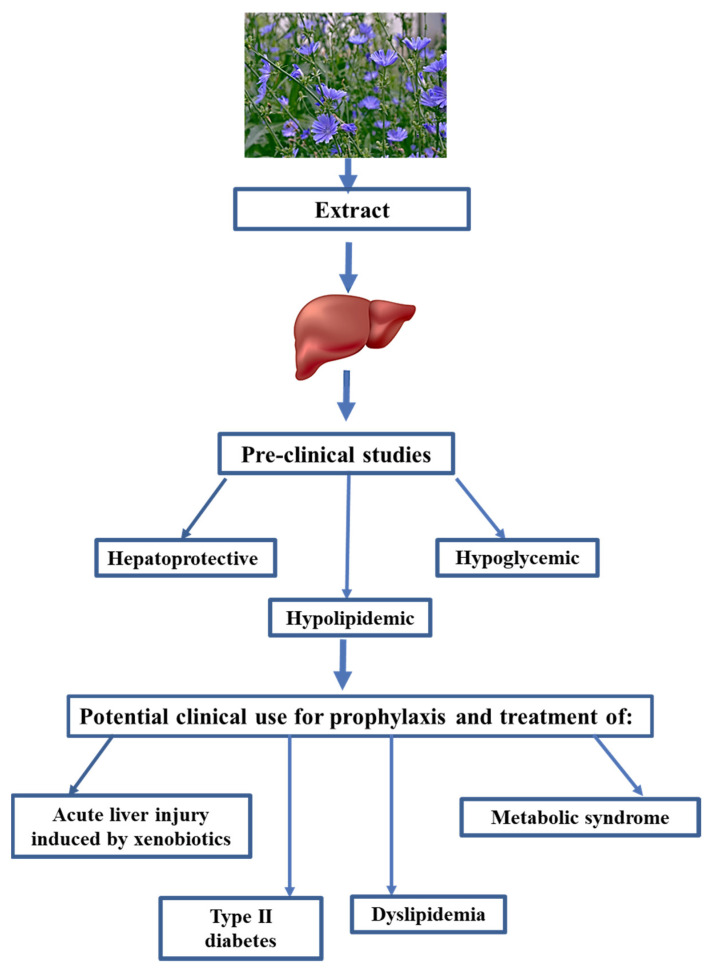
Preclinical properties of chicory and potential clinical use.

**Table 1 pharmaceuticals-16-01471-t001:** The effect of CE on body weight, relative liver weight, and biochemical serum profile of rats with acute liver injury.

Biomarker (Mean ± SD)	Treatment Groups ^a^			
	Untreated Control	HgCl_2_-Treated	HgCl_2_-Treated + CE 100 mg/kg	HgCl_2_-Treated + CE 500 mg/kg
Body weight on day 10 after experiment initiation (% of initial body weight)	131.6 ± 1.7	112.4 ± 2.4 *	117.1 ± 2.6	120.7 ± 2.9 *
Body weight on day 21 after experiment initiation (% of initial body weight)	147.5 ± 2.5	140.1 ± 1.9	144.9 ± 1.1	145.9 ± 2.0
Relative liver weight	3.93 ± 0.13	5.12 ± 0.15 *	4.53 ± 0.16 #	4.16 ± 0.06 #
Total protein, g/L	74.7 ± 1.0	83.7 ± 1.1 *	75.3 ± 0.7 #	76.3 ± 0.9 #
Glucose, mmol/L	7.04 ± 0.17	9.01 ± 0.16 *	7.2 ± 0.14 #	7.5 ± 0.13 #
Total cholesterol, mmol/L	1.72 ± 0.04	2.80 ± 0.08 *	2.1 ± 0.08 #	1.8 ± 0.05 #
Triglycerides, mmol/L	0.98 ± 0.11	1.55 ± 0.12 *	1.2 ± 0.15 #	1.2 ± 0.15 #
Total bilirubin, mmol/L	4.9 ± 0.2	7.8 ± 0.1 *	4.6 ± 0.1 #	4.8 ± 0.1 #
γ-glutamyl transferase (GGT), U/L	6.5 ± 0.7	11.0 ± 04 *	9.0 ± 0.5 #	8.7 ± 0.7 #
Alkaline phosphatase (ALP), U/L	784.0 ± 28.4	1005 ± 44.2 *	971.0 ± 80.4	897.6 ± 57.0 #
Aspartate aminotransferase (AST), U/L	136.8 ± 4.2	175.1 ± 4.4 *	155 ± 10.5	151.1 ± 7.1 #
Alanine aminotransferase (ALT), U/L	89.3 ± 3.7	126.3 ± 4.1 *	90.0 ± 5.4 #	80.9 ± 1.6 #

^a^—Hepatoprotective activity of CE was evaluated at 21 days after HgCl_2_-induced liver damage; *—Significantly different compared with control, *p* < 0.05; #—Significantly different compared with HgCl_2_-treated group, *p* < 0.05.

**Table 2 pharmaceuticals-16-01471-t002:** Effect of CE on the serum biochemical profile of rats treated with a single i.p. injection of Tween-80.

Biomarkers (Mean ± SD)	Treatment Groups ^a^			
	Untreated Control	Tween-80-Treated	Tween-80-Treated + CE 100 mg/kg	Tween-80-Treated + CE 500 mg/kg
Total cholesterol, mmol/L	1.56 ± 0.09	1.83 ± 0.10	1.62 ± 0.08	1.74 ± 0.05
High-density lipoproteins (HDL), mmol/L	0.73 ± 0.03	0.57 ± 0.03 *	0.65 ± 0.04	0.67 ± 0.03 #
Low-density lipoproteins (LDL), mmol/L	0.58 ± 0.04	0.84 ± 0.03 *	0.57 ± 0.02 #	0.62 ± 0.04 #
Triglycerides, mmol/L	0.68 ± 0.03	0.95 ± 0.04 *	0.60 ± 0.07 #	0.65 ± 0.06 #
Glucose, mmol/L	5.23 ± 0.21	5.95 ± 0.13 *	4.76 ± 0.17 #	4.47 ± 0.15 #

^a^—Hypolipidemic and hypoglycemic activities of CE were evaluated 8 h post Tween-80 administration; *—Significantly different compared with control, *p* < 0.05; #—Significantly different compared with Tween-80-treated group, *p* < 0.05.

**Table 3 pharmaceuticals-16-01471-t003:** Effect of CE on body weight and serum biochemical profile of rats with alimentary hyperlipidemia.

Biomarkers (Mean ± SD)	Treatment Groups ^a^			
	Untreated Control	Cholesterol–Mercazolil Treatment	Cholesterol– Mercazolil Treatment + CE 100 mg/kg	Cholesterol– Mercazolil Treatment + CE 500 mg/kg
Body weight on day 29 after experiment initiation (% of initial body weight)	119.4 ± 1.9	123.1 ± 2.4	119.2 ± 1.7	115.0 ± 1.5 #
Total cholesterol, mmol/L	1.65 ± 0.08	1.82 ± 0.04 *	1.56 ± 0.07 #	1.53 ± 0.05 #
High-density lipoproteins (HDL), mmol/L	0.80 ± 0.04	0.93 ± 0.03	0.83 ± 0.02	0.77 ± 0.03
Low-density lipoproteins (LDL), mmol/L	0.55 ± 0.03	0.53 ± 0.05	0.38 ± 0.04 #	0.39 ± 0.03 #
Triglycerides, mmol/L	0.76 ± 0.06	0.99 ± 0.08 *	0.69 ± 0.04 #	0.78 ± 0.02 #
Glucose, mmol/L	4.80 ± 0.14	5.04 ± 0.13	5.01 ± 0.17	5.14 ± 0.18

^a^—Hypolipidemic activities of CE were evaluated 29 days after cholesterol–mercazolil administration with or without CE treatment; *—Significantly different compared with control, *p* < 0.05; #—Significantly different compared with the alimentary hyperlipidemia group, *p* < 0.05.

## Data Availability

Data is contained within the article and Appendix A.

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
