# Peer review of "Modulation of Hepatic Functions by Chicory (Cichorium intybus L.) Extract: Preclinical Study in Ratsâ€"

_pharmaceuticals, 2023, doi:10.3390/ph16101471_

Round 1
Reviewer 1 Report
I reviewed the manuscript titled “Modulation of hepatic functions by chicory (Cichorium intybus, L) extract: pre-clinical study in rats #
Remove the comma and full stop from the tile
1-4 authors' affiliation is the same. It is recommended to use only one and represent with superscript 1 after the name.
Line 20: Presented herein study investigated.. can be revised as Therefore, the STUDY…
Line 65 and elsewhere: compounds name should not be in capital letters (starting word)
Provide the ref for section 4.2. Profiling of Polyphenolic compounds present in CE.
Provide the ref for section 4.3. Quantification of total and dominant polyphenols in the CE
Provide citation for section 4.6. Analysis of the CE hypolipidemic effect on the model of hyperlipidemia caused by a single 324 intraperitoneal injection of Tween-80.
Line 89: remove full stop before figure
Most of the sections in the results are well presented.
The discussion is very poor. Authors must provide in-depth discussion along with citations
Figure 3 quality is extremely poor
Figure 1: quality must be improved. Different numbers must be denoted in figure caption
Conclusions should be revised to reflect the findings. As such, it is not clear for the readers. Readers also do not know what was in previous research. Authors should make conclusions based on the study findings
References are not according to the journal format
Overall, the manuscript has quality but authors must consider above suggestions in order to further improve the manuscript quality
Author Response
Reviewer 1.
Comments and Suggestions for Authors
I reviewed the manuscript titled “Modulation of hepatic functions by chicory (Cichorium intybus, L) extract: pre-clinical study in rats #
- Remove the comma and full stop from the tile.
- Corrected.
- 1-4 authors' affiliation is the same. It is recommended to use only one and represent with superscript 1 after the name.
- Corrected.
- Line 20: Presented herein study investigated. can be revised as Therefore, the STUDY…
- This has been edited to improve readability.
- Line 65 and elsewhere: compounds name should not be in capital letters (starting word)
- Thank you, bioactive compound names were edited according to the recommendations.
- Provide the ref for section 4.2. Profiling of Polyphenolic compounds present in CE.
- Appropriate references were added
6. Provide the ref for section 4.3. Quantification of total and dominant polyphenols in the CE
- Appropriate reference was added
- Provide citation for section 4.6. Analysis of the CE hypolipidemic effect on the model of hyperlipidemia caused by a single intraperitoneal injection of Tween-80.
- Appropriate reference was added
- Line 89: remove full stop before figure
- Removed. Thank you for catching it.
- Most of the sections in the results are well presented.
- Thank you for your appreciation of the data presentation.
10. The discussion is very poor. Authors must provide in-depth discussion along with citations.
- We have significantly expanded the Discussion by adding more in-depth analyses of available literature and related citations. Please see the revised manuscripts.
- Figure 3 quality is extremely poor
- The quality of the figure has been improved.
- Figure 1: quality must be improved. Different numbers must be denoted in figure caption
- The quality of the figure has been improved for Figure 1S (former Figure 1). We have replaced the figure’s chromatogram and numbered dominated bioactive compounds. Instead of adding descriptions of more than 20 compound names and not duplicating information, the peaks’ numbers were added to the supplemental table 1.
13. Conclusions should be revised to reflect the findings. As such, it is not clear for the readers. Readers also do not know what was in previous research. Authors should make conclusions based on the study findings
- Conclusions were edited.
- References are not according to the journal format
- references were formatted with Chicago-mdpi style
Overall, the manuscript has quality but authors must consider above suggestions in order to further improve the manuscript quality
- Thank you for your helpful suggestions to improve the quality of the manuscript.
Reviewer 2 Report
1. After mentioning the plant full name Cichorium intybus L you must then use its abbreviated name C. intybus throughout the whole manuscript.
2. All chromatograms must be added to the supplementary material.
3. Replace the old technique not accurate (High Performance Liquid Chromatography (HPLC)) into LC-MS-MS.
4. Your discussion and conclusion parts need more improvements.
5. The whole manuscript needs major grammars, typos and editing corrections.
6. Add histological figure of the liver for control and treated groups
manuscript needs major grammars, typos and editing corrections.
Author Response
- After mentioning the plant full name Cichorium intybus L you must then use its abbreviated name C. intybus throughout the whole manuscript.
- Thank you for catching it. It was corrected per the reviewer's recommendations
- All chromatograms must be added to the supplementary material.
- Table 1 and figure1 are presented as supplemental materials
- Replace the old technique not accurate (High-Performance Liquid Chromatography (HPLC)) into LC-MS-MS.
- Edited based on the current designation of HPLC and LC-MS/MS
- Your discussion and conclusion parts need more improvements.
- We have significantly expanded the Discussion by adding more in-depth analyses of available literature and related citations. The conclusion has also been edited.
- The whole manuscript needs major grammars, typos and editing corrections.
- The manuscript has been edited and reviewed by the authors including a native English speaker
- Add histological figure of the liver for control and treated groups
- figure 2 (former figure 3) has the requested information. Figure 2A- histology liver of the control animal; figure 2B – liver of animals treated with cholesterol-mercazolil and figure 2C-D are livers of rats treated with cholesterol-mercazolil and CE at different doses. The figure legend was edited to clarify the information.
Reviewer 3 Report
This manuscript is seriously flawed and unscientific proof of concept. The authors make an approach to the state of the art absolutely inadequate, being a simple disclosure. The study is not justified and does not contribute novelty. The methodological issues are described in a very light way and do not allow this manuscript to be reproducible. The authors do not justify the sample size. The results are presented in an erratic way and without an appropriate causal sequence. Figure 3 is of very poor quality, nothing is described. The discussion is an intention, with outdated references. The conclusions are not supported by the results. In addition, the text of the manuscript uses inappropriate English, it looks like an internet translation.
English very difficult to understand/incomprehensible
Author Response
- This manuscript is seriously flawed and unscientific proof of concept.
- Unfortunately, we disagree with the reviewer. The study methods used well-known proof of concepts for the study design, data collection, and analyses. These are well-recognized within the scientific community and used by multiple researchers in this area of research and investigations. Throughout the manuscript with have referenced methods used in the study that have used an approach and methods such as that outlined in this manuscript.
- The authors make an approach to the state of the art absolutely inadequate, being a simple disclosure. The study is not justified and does not contribute novelty.
- Again, we disagree with the reviewer, this study presents data demonstrating that the original extract from the aerial part of chicory possesses simultaneously properties that include hepatoprotective, hypolipidemic, and hypoglycemic. To the best of our knowledge, our study is novel and within the published literature there is a paucity of data demonstrating multi-targeted properties associated with the extract from the aerial part of chicory. Most often, published studies present only one activity of the extract from the aerial part or roots of chicory.
- The methodological issues are described in a very light way and do not allow this manuscript to be reproducible.
- As experienced researchers within this area, we disagree and believe that the study methods are described adequately and provide enough details for other investigators to undertake confirmatory and/or reproducibility experiments to confirm this study finding(s). The methods and analytical approaches used within the experimental design of this study are well-known within pharmacology/toxicology. We have added additional references to previously published studies that have used the same methods and analysis approaches.
- The authors do not justify the sample size. The results are presented in an erratic way and without an appropriate causal sequence.
- Unfortunately, we disagree with the reviewer that we have not justified the sample size. The sample size and study design were guided by the recommendation of the Board of the Eurasian Economic Commission dated May 21, 2020, #10 “Guidelines for Conducting Preclinical Toxicity Studies with Repeated (Multiple) Administration of Active Ingredients of Medicinal Products for Medical Use”, the "European Convention for the Protection of Vertebrate Animals Used for Experimental and other Scientific Purposes (ETS 123)," and the "Guidelines for conducting pre-clinical studies of drugs" implemented by the Ministry of Public Health of the Russian Federation” were used to determine the number of experimental animals in the group. These documents regulate a sufficient number of animals for statistical analysis of the results and at the same time compliance with the principles of 3R and state that the size of the experimental group should be sufficient to carry out a full scientific interpretation of the data obtained. We also disagree that the results are presented in an erratic way, the results are presented in the logical context of how the experiments were conducted. We have made edits as suggested by other reviewers that include adding additional references and clarity to figure descriptions.
5. Figure 3 is of very poor quality, nothing is described.
- As suggested by another reviewer we have updated figure 3, which is now figure 2. Figure 2A- histology liver of the control animal; figure 2B – liver of animals treated with cholesterol-mercazolil and figure 2C-D are livers of rats treated with cholesterol-mercazolil and CE at different doses. The figure legend was edited to clarify information.
6. The discussion is an intention, with outdated references.
- Unfortunately, I (EYE) disagree with the reviewer regarding outdated references. There are a few references dated by 2014 and 2015 years; the use of these references was justified. The remaining references are dated 2020-2023. We have expanded the discussion section pre reviewers 1 and 2 requested.
7. The conclusions are not supported by the results.
We disagree that the conclusions are not supported by the results, as suggested by another reviewer we have expanded the conclusions.
8. In addition, the text of the manuscript uses inappropriate English, it looks like an internet translation.
- The manuscript has been edited and reviewed by the authors including a native English speaker
Reviewer 4 Report
Comment 1- In the introduction, I suggest writing a brief discussion of the mechanism of drug overdose-induced liver injury.
Comment 2- For the profiling of polyphenolic compounds present in CE, C-18 columns were used. However, it is my personal experience that the pump pressure in the C-18 column increases, which would interfere with the results. If the authors faced the same situation, how did they manage?
Comment 3- Isoflurane anaesthesia induces liver injury by regulating the expression of insulin-like growth factor. Did the authors standardised the safe doses before providing anaesthesia to rats?
Comment 4- The authors used mercazolil tabs for analysis of total cholesterol and triglycerides levels along with cholesterol in experimental rats. It is my suggestion to discuss its significance in current study and its mechanism of action.
Comment 5- Introduction may be improved adding new information in order to provide an adequate state-of-the-art.
Hesperidin, a Bioflavonoid in Cancer Therapy: A Review for a Mechanism of Action through the Modulation of Cell Signaling Pathways. Molecules. 2023; 28(13):5152. https://doi.org/10.3390/molecules28135152
Comment 6- In the introduction I suggest writing a brief paragraph regarding the toxicity and nutritional assessment of extract of chicory (Cichorium intybus 2 L.) extract.
Comment 7- Recent studies have revealed that inflammation is generally implicated in the liver diseases ranging from the initial to the late stage and is a common pathophysiological response to liver injury.
Cite the following article.
6-Gingerol, a Major Ingredient of Ginger Attenuates Diethylnitrosamine-Induced Liver Injury in Rats through the Modulation of Oxidative Stress and Anti-Inflammatory Activity.
Comment 8- The authors are advised to improve the figures.
Comment 9- English grammar and spelling check should be corrected in the whole manuscript.
Comment 10- The conclusions are too few compared to the extensive discussion in the text. I think they should be expanded. The article’s conclusion should include state of the field of science to date, with a brief description of achievements and shortcomings.
Comment 11- I think it is necessary to add this work's critical achievements at the very beginning of discussion. It is also desirable to combine parts of the work with a suggestion of why it works that way.
Comment 12- Recommendations: Please review your finding in the figure (or graphical abstract) with a brief description and value of found effects.
English grammar and spelling check should be corrected in the whole manuscript.
Author Response
Reviewer 4.
Dear reviewer, thank you for the time that you have found to review our paper. The comments raised by the reviewer have been answered/justified and listed point-by-point below.
Comment 1- In the introduction, I suggest writing a brief discussion of the mechanism of drug overdose-induced liver injury.
Response 1. Our manuscript was not intended to study the mechanisms responsible for the hepatoprotective properties of a novel chicory extract. We feel that the Discussion of the mechanisms of DILI in the Introduction section is not justified. However, per the reviewer's request, we added a sentence stating the basic mechanisms responsible for liver injury following DILI and cited it.
Comment 2- For the profiling of polyphenolic compounds present in CE, C-18 columns were used. However, it is my personal experience that the pump pressure in the C-18 column increases, which would interfere with the results. If the authors faced the same situation, how did they manage?
Response 2. We used a Luna 5µm C18 100 Å (250 x 4.6 mm) column (Phenomenex, CA). During the analysis, the pressure in the column did not increase or interfere with obtaining the results.
Comment 3- Isoflurane anaesthesia induces liver injury by regulating the expression of insulin-like growth factor. Did the authors standardised the safe doses before providing anaesthesia to rats?
Response 3. Indeed, there are several publications discussing an IGF-1 role in isoflurane liver injury. It appears that this injury takes place following prolonged exposure (90 min) to isoflurane (see article: Exp Ther Med. 2017 Apr; 13(4): 1608–1613. doi: 10.3892/etm.2017.4157). Our animals, including control animals, were exposed to isoflurane for 30-45 seconds, wherefore all rats had the same liver injury, if any.
Comment 4- The authors used mercazolil tabs for analysis of total cholesterol and triglycerides levels along with cholesterol in experimental rats. It is my suggestion to discuss its significance in current study and its mechanism of action.
Response 4. The significance of the current study is in presenting a novel extract from the aerial part of wild chicory that possesses simultaneously hepatoprotective, hypolipidemic, and hypoglycemic properties. CE hypolipidemic properties were tested on two hyperlipidemia modes. One of them is the alimentary hyperlipidemia, which uses mercazolil. The mercazolil model supports the finding of the Tween model. The merit of using the mercazolil model is added to the Discussion section.
Comment 5- Introduction may be improved adding new information in order to provide an adequate state-of-the-art.
Hesperidin, a Bioflavonoid in Cancer Therapy: A Review for a Mechanism of Action through the Modulation of Cell Signaling Pathways. Molecules. 2023; 28(13):5152. https://doi.org/10.3390/molecules28135152
Response 5. The introduction section references old as well as recently published articles to adequately introduce the study results. The majority of publications references are from 2019-2022. We are confused as to why the reviewer asked us to reference the "hesperidin" paper. CE does not contain this constituent. Moreover, our manuscript does not discuss cancer therapy.
Comment 6- In the introduction I suggest writing a brief paragraph regarding the toxicity and nutritional assessment of extract of chicory (Cichorium intybus 2 L.) extract.
Response 6. We have reported an acute toxicity study.The information about acute CE toxicity will be added to the Introduction. No nutritional assessment will be added. This is a potential drug, not a nutraceutical or dietary supplement; therefore, this information is irrelevant to the study.
Comment 7- Recent studies have revealed that inflammation is generally implicated in the liver diseases ranging from the initial to the late stage and is a common pathophysiological response to liver injury.
Cite the following article.
6-Gingerol, a Major Ingredient of Ginger Attenuates Diethylnitrosamine-Induced Liver Injury in Rats through the Modulation of Oxidative Stress and Anti-Inflammatory Activity.
Response 7. Indeed, the studies investigating the mechanisms of the DILIs attribute the protective properties of the anti-oxidant and anti-inflammatory properties of these molecules. Diethylnitrosamine is a known hepatocarcinogen. We have added the recommended reference to the Discussion.
Comment 8- The authors are advised to improve the figures.
Response 8. Figures were improved.
Comment 9- English grammar and spelling check should be corrected in the whole manuscript.
Response 9. The manuscript was checked by a native speaker.
Round 2
Reviewer 2 Report
The authors did all the required corrections and I have not any more comments
Author Response
Thank you!
Reviewer 3 Report
This manuscript is a revised version of pharmaceuticals-2452707. The authors have not made any significant changes. The authors have confined themselves to minor aspects of style. This manuscript does not support the conclusions with the results presented. This manuscript is not fit for publication.
English very difficult to understand/incomprehensible
Author Response
The statements provided by Reviewer 3 were very general, with no specifics and no particular guidelines on what they wanted to be edited or changed.
Reviewer 3 made the following statements after we had submitted a revised version of the manuscript and responded to the original comments.
- “ The authors have not made any significant changes”. Unfortunately, because the comments from Reviewer 3 were not specific about what they wanted us to do and did not explain what deficiencies the manuscript had, it was extremely hard for us to address. They stated our manuscript was flawed but did not expand on how it was flawed. They stated the approach was inadequate and not justified but did not make any specific examples of what needed to be improved. In contrast, the other two reviewers who had provided specifics of what they wanted edited or changed were satisfied with the edits and revised manuscript provided by the authors.
- Reviewer 3 stated, “This manuscript does not support the conclusions with the results presented.” Again, there was no description given, in contrast, Reviewer 1 provided suggestions for the conclusion, which were undertaken by the authors in the revised version of the manuscript. Our conclusions stated that “This study's results provide data demonstrating that chicory (Cichorium intybus L.) extract from aerial parts of the plant possesses hepatoprotective, hypolipidemic, and hypoglycemic properties in an investigation undertaken using a mouse model. Unfortunately, little is known about the effects of chicory extracts on human organisms. Therefore, it is imperative that clinical trials are designed that can evaluate chicory extracts and determine their efficacy and toxicity in humans”. Our study demonstrated, using the model of acute hepatotoxicity, that chicory extract has hepatoprotective properties by improving key liver functions. The following two models of hyperlipidemia demonstrated that the treatment with chicory extract could improve lipid profile and glucose levels in experimental rats. Therefore, we felt that the conclusions that we outlined from the study were justified based on the data and results that were obtained, and this was accepted by the other two reviewers.
- Reviewer 3 stated “The authors have confined themselves to minor aspects of style”. We are not sure what exactly this refers to; we did update figures as requested by the reviewer. We also undertook other edits and suggestions as requested by reviewers 1 and 2. As reviewer 3 did not provide any specifics in their original comments, it was difficult for us to provide any additional responses to them.
4, Reviewer 3 stated, “This manuscript is not fit for publication”. This is a blank statement, which was not supported with any details of why and what is so wrong with the manuscript. Reviewer 3 did not provide any details or justification for making this statement, and in contrast, Reviewers 1 and 2 did not feel the same way and were happy with the revisions that were undertaken. Reviewer 3 did not, in the original request for revisions or in the subsequent responses, explain why the manuscript was not fit to be published in the special issue. We feel we were left in an impossible situation as the initial feedback from reviewer 3 was also so general and did not provide us any guidance on what to change or explain. For us, it comes across as the reviewer did not like the manuscript right from the start.